# TESTJUDGE: A RIGOROUS BENCHMARK FOR UNIT TEST GENERATION AND QUALITY ASSESSMENT

## ABSTRACT

Test generation is a critical component of automated code generation, yet existing benchmarks primarily evaluate generated tests using pass rates, overlooking test comprehensiveness and error-detection capabilities. We introduce TestJudge, a benchmark designed to evaluate both the quality and error-detection capabilities of generated unit tests. TestJudge contains 8,000 programming problems in Python and C++ sourced from Codeforces. For each problem, we provide 10 diverse code submissions with known correctness labels, where a generated test is considered valid only if it correctly classifies all 10 submissions according to ground-truth verdicts. Our evaluation of 13 state-of-the-art models using verdict matching rate and coverage metrics reveals significant challenges in current approaches. The best-performing model, Gemini-2.5-Pro, achieves verdict matching rates of only 59.75% for Python and 11.50% for C++. Notably, we observe a striking performance gap when comparing test generation versus direct problem-solving tasks on identical problems, with problem-solving success rates being considerably higher. This discrepancy suggests that models may rely on problem memorization rather than developing robust testing strategies, highlighting a critical limitation in current automated test generation approaches.

## 1 INTRODUCTION

Software testing is a crucial component of software development, as it ensures that functional requirements are met and identifies potential defects. However, writing high-quality test cases is time-consuming and costly, accounting for roughly 15% of the overall development effort (Daka & Fraser, 2014).

The recent success of large language models (LLMs) in solving programming problems (Chen et al., 2021; Austin et al., 2021) has sparked a natural question: if LLMs can solve problems, can they also generate high-quality test cases? After all, a model that understands the problem well enough to solve it should also be capable of designing tests to validate solutions.

To investigate this hypothesis, we conducted a systematic evaluation and uncovered a surprising result. LLMs perform well at solving programming tasks but struggle to generate effective test cases. This issue is particularly severe for C++ problems, where the best-performing model achieves only 11.50% verdict matching rate. A closer look at failure cases reveals that the majority are not grossly incorrect, but rather involve subtle reasoning errors, such as missing corner cases or implicit constraints.

Unfortunately, most existing benchmarks for automated test generation remain flawed in evaluating test generation capabilities. Prior work commonly reports pass rate and coverage (Wang et al., 2024; 2025; Xu et al., 2025); some benchmarks also report mutation score (Zhang et al., 2024; Jain et al., 2024). Pass rate and coverage measure only correctness and code coverage, not a model's ability to detect errors. Mutation score targets coarse changes in code blocks and misses subtle logical errors. **As a result, current evaluations may overestimate LLMs' true ability to ensure solution robustness.**

Motivated by this gap, we propose TestJudge, a benchmark designed to measure LLMs' ability to detect subtle logical errors. TestJudge contains 800 programming problems from the Codeforces online competition platform. **To reduce randomness in the evaluation, each problem is paired**

**with 10 real code submissions.** To assess generalization across languages, half of the problems are in Python and half in C++. We introduce two metrics: **Verdict Matching Rate**: whether LLM-generated tests agree with the ground-truth Codeforces verdicts. **Coverage**: how thoroughly the tests exercise different code paths.

Our evaluation across both commercial and open-source LLMs shows that even the strongest model, Gemini-2.5-Pro (Google DeepMind, 2025), achieves only 59.75% matching on Python and performs drastically worse on C++. Revisiting these results, we highlight a key insight: **LLMs may rely more on memorization than on genuine reasoning or problem understanding.** They can reproduce correct solutions, yet fail to generate tests that expose subtle logical errors and hidden corner cases. We refer to this phenomenon as the **Problem Memorization** - the disconnect between problem-solving ability and test-generation ability - which TestJudge is designed to reveal and ultimately help address.

Our work makes the following contributions:

- **Benchmark.** We release a benchmark focused on evaluating the capability of LLMs in detecting subtle logical errors. The dataset consists of 8000 codes from 800 programming competition questions in Python and C++.
- **Evaluation.** We evaluate 13 state-of-the-art models on our benchmark and find that LLMs struggle to comprehensively evaluate code submissions.
- **Analysis.** We compare execution logs from problem solving and test generation. And we find that LLMs tend to memorize solution procedures rather than understand the problems.

## 2 TESTJUDGE

In this section, we introduce TestJudge from several perspectives. The overview of TestJudge is shown in Figure 1. We describe the construction of the dataset (Section 2.1), the tasks in our benchmark (Section 2.2), and the evaluation metrics (Section 2.3). Finally, we summarize the properties of TestJudge (Section 2.4).

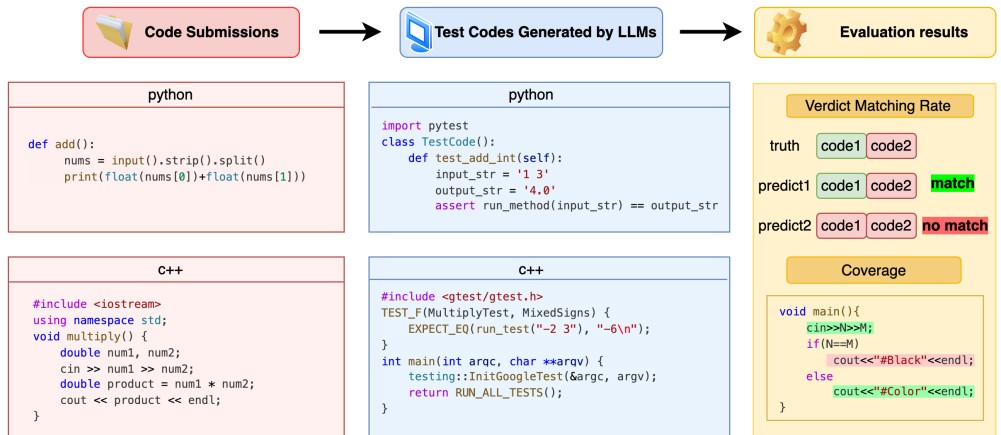

Figure 1: Overview of TestJudge. We provide multiple code submissions for each programming problem and prompt LLMs to generate tests. We then execute these tests to obtain evaluation metrics, including verdict matching rate, and coverage.

### 2.1 BENCHMARK CONSTRUCTION

**Data collection.** We begin with open-source datasets containing code submissions on CodeForces [1,2], and then retrieve missing fields for each submission from the Codeforces platform[3]. For each

---

[1]https://huggingface.co/datasets/MatrixStudio/Codeforces-Python-Submissions

[2]https://sites.google.com/site/miningprogcodeforces/home/dataset

[3]https://codeforces.com

submission, the JSON record contains the problem description, input specification, output specification, sample input, sample output, verdict, and source code.

**Data filtering.** We filter out submission records containing incomplete fields. Then, we prompt GPT-4.1 and Doubao-1.5-Pro-32K to filter out unsolvable problems as well as those with unclear statements.

**Data extraction.** We construct tests based on sample input and sample output, run them in the execution environment, and filter out submission records that fail. We then select 800 problems, each paired with 10 code submissions-400 in Python and 400 in C++.

We ultimately select the dataset consisting of 8,000 code submissions as the evaluation set for Test-Judge. The problem tags and sources are provided in Appendix A.

## 2.2 TASK DESCRIPTION

We propose two tasks: (1) test generation, and (2) problem solving. The detailed procedure is shown in Figure 2. Orange indicates test generation, and purple indicates problem solving.

**(1) Test generation.** We provide LLMs with the problem description, input–output specifications, and input–output samples, ten code submissions with known correctness labels, and prompt them to generate multiple test cases (see Appendix B for full prompts). We then extract the last code block from the response. We compile the code and enforce a 30-second time limit. We run the extracted code separately from the code submissions to produce test logs and metrics. If compilation fails, the run times out, or an unknown error occurs, we mark the case as a a matching failure.

**(2) Problem solving.** We prompt LLMs to generate solutions for the same problems in test generation. Then, we execute each solution with the official test suite. Finally, we compute the success rate of the solution (Pass@1) and compare it to the verdict matching rate in the test-generation task. In this task, LLMs generate incorrect test cases, which means that they do not fully understand the problem. A high success rate of solutions on the same problems indicates the phenomenon of problem memorization. This procedure provides a new way to quantify problem memorization in LLMs.

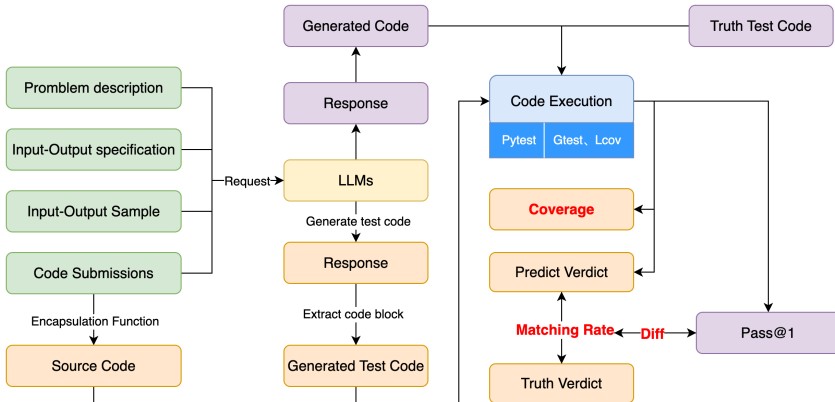

Figure 2: Schematic illustration of the two tasks (test generation and problem solving). The test-generation task, highlighted in orange, involves guiding LLMs to create test code, while the problem-solving task, shown in purple, focuses on generating solutions and evaluating them with an official test suite.

## 2.3 METRICS

We introduce three evaluation metrics for TestJudge, including the verdict matching rate, coverage, and the difference between the success rate of problem-solving (Pass@1) and the verdict matching rate.

**Verdict Matching Rate (VMR).** Each sample contains one problem and ten code submissions. We apply the LLM-generated test code to the ten code submissions. If all ten verdicts match the ground-truth verdicts, we label the sample as a match. We select the proportion of matched samples among all samples as our metric. The formula can be expressed as follows:

$$\text{Verdict Matching Rate} = \frac{\text{Number of matched samples}}{\text{Total number of samples}} \times 100\% \tag{1}$$

**Coverage.** Coverage is a standard unit-testing metric. Coverage measures the proportion of code lines executed during testing out of the total lines of code.

**Difference between the success rate of problem-solving and VMR (Diff).** For each problem, we prompt the LLM to produce a solution and compute the success rate of problem-solving (Pass@1). Then, we use the difference between Pass@1 and VMR to quantify the severity of problem memorization.

## 2.4 PROPERTIES OF TESTJUDGE

Table 1: Comparison between TestJudge and existing test generation benchmarks. TestJudge is the only benchmark for testing the capability to detect subtle logical errors.

| Dataset | Size | Multiple language | Coverage | Case legality | Logic correction |
|---|---|---|---|---|---|
| TESTBENCH | 108 | ✗ | ✔ | ✔ | ✗ |
| TESTEVAL | 210 | ✗ | ✔ | ✗ | ✗ |
| TESTGENEVAL | 1210 | ✗ | ✔ | ✔ | ✗ |
| PROJECTTEST | 60 | ✔ | ✔ | ✔ | ✗ |
| CLOVER | 845 | ✗ | ✔ | ✗ | ✗ |
| TESTJUDGE | 800 | ✔ | ✔ | ✔ | ✔ |

To provide a deeper understanding of the distinctions among these benchmarks, we provide a detailed explanation of each property, including multiple-language support, line coverage, case legality, and logic correction, in Table 1.

**Multiple languages.** There are multiple programming languages in practice. The benchmarks that include only a single language have inherent limitations. TestJudge considers the evaluation of Python and C++.

**Line coverage.** This metric is a standard metric in unit testing. Coverage assesses comprehensiveness of test cases.

**Case legality.** Existing benchmarks often report the pass rate of generated tests. However, passing all generated cases shows that these cases meet the requirements, but does not necessarily demonstrate comprehensive coverage. TestJudge measures legality as the proportion of correct programs accepted by the generated tests.

**Logic correction.** Existing benchmarks predominantly use mutation scores to evaluate the error-detection capability of LLMs, but they primarily inject macro errors at the code level, such as flipping logical operators, renaming variables, or deleting code. In practice, it is common for solutions to not fully cover the problem, resulting in subtle logical errors. **TestJudge is the first benchmark to evaluate logical error detection using multiple code submissions per problem.**

## 3 EXPERIMENTAL EVALUATION

### 3.1 EXPERIMENTAL SETUP

We evaluated 13 popular LLMs, including both commercial and open-source models (see Appendix C). We set `temperature` and `top_p` to 0.8. We set `max_tokens` to 1,024 for non-thinking models and to 16,384 for thinking models. We used identical parameters across models to ensure a fair comparison. We computed Python metrics using `Pytest` and C++ metrics using `Gtest` and `Lcov`.

## 3.2 TEST GENERATION

Table 2: Metrics of various LLMs for test generation. The best results are bolded.

| Model | Python VMR | C++ VMR | Python Coverage | C++ Coverage |
|---|---|---|---|---|
| **Non-thinking Models** | | | | |
| Doubao-1.5-Pro-32k-250115 | 17.25% | 1.50% | 97.28% | 90.13% |
| Qwen3-235B-A22B | 15.50% | 2.00% | 97.87% | 90.86% |
| GPT-4o | 11.75% | 0.00% | 97.13% | 90.88% |
| GPT-4.1 | 19.00% | 1.75% | 97.41% | 90.85% |
| DeepSeek-V3 | 12.75% | 1.50% | 97.43% | 90.68% |
| Claude-3.7-Sonnet-20250219 | 19.50% | 1.50% | 97.81% | 91.18% |
| Claude-4-Sonnet-20250514 | 25.25% | 3.75% | 97.69% | 91.24% |
| **Thinking Models** | | | | |
| Doubao-1.5-thinking-Pro-250415 | 47.75% | 2.00% | 97.67% | 91.08% |
| Doubao-Seed-1.6-thinking-250615 | 53.00% | 6.75% | 97.35% | 90.37% |
| Claude-3.7-Sonnet-thinking-20250219 | 27.50% | 4.00% | 97.81% | **91.72%** |
| Claude-4-Sonnet-thinking-20250514 | 36.00% | 3.50% | 97.79% | **91.72%** |
| DeepSeek-R1 | 39.00% | 3.50% | **97.98%** | 90.55% |
| Gemini-2.5-Pro | **59.75%** | **11.50%** | 97.90% | 91.14% |

Table 2 reports the Verdict Matching Rate (VMR) and Coverage for each model in the test generation. From the table, we observations three points.

(1) The average VMR of non-thinking models is lower than that of thinking models. Genmini-2.5-Pro performs best, yet its VMR is only 59.75% in Python and 11.50% in C++, indicating that **LLMs struggle to generate effective test code**.

(2) For the same model, metrics vary by programming language. **VMR and Coverage are much lower in C++ than in Python**, indicating stronger capability in Python. This pattern is consistent with the results of other multilingual test benchmarks (Zan et al., 2025; Yang et al., 2024).

(3) From a practical perspective, higher line coverage exercises more execution paths and makes validation more complete, which tends to increase VMR. Table 2 shows **a positive correlation between coverage and VMR**, which is consistent with the empirical observations.

We further analyze the logs of unmatched cases to explain failures. See Section 4.1.1.

## 3.3 TEST GENERATION VS PROBLEM SOLVING

Table 3: Comparison of test-generation task and problem-solving task. The difference between Pass@1 and VMR is in parentheses. The biggest difference is bolded.

| Model | Python VMR | Python Pass@1 | C++ VMR | C++ Pass@1 |
|---|---|---|---|---|
| **Non-thinking Models** | | | | |
| Doubao-1.5-Pro-32k-250115 | 17.25% | 80.25(+63.00)% | 1.50% | 43.25(+41.75)% |
| Qwen3-235B-A22B | 15.50% | 81.25(+65.75)% | 2.00% | 52.75(+50.75)% |
| GPT-4o | 11.75% | 61.00(+49.25)% | 0.00% | 42.00(+42.00)% |
| GPT-4.1 | 19.00% | 82.25(+63.25)% | 1.75% | 47.75(+46.00)% |
| DeepSeek-V3 | 12.75% | 90.25(**+77.50**)% | 1.50% | 53.50(+52.00)% |
| Claude-3.7-Sonnet-20250219 | 19.50% | 74.25(+54.75)% | 1.50% | 49.50(+48.00)% |
| Claude-4-Sonnet-20250514 | 25.25% | 84.50(+59.25)% | 3.75% | 61.50(+57.75)% |
| **Thinking Models** | | | | |
| Doubao-1.5-thinking-Pro-250415 | 47.75% | 85.75(+38.00)% | 2.00% | 62.75(+60.75)% |
| Doubao-Seed-1.6-thinking-250615 | 53.00% | 84.00(+31.00)% | 6.75% | 51.75(+45.00)% |
| Claude-3.7-Sonnet-20250219-thinking | 27.50% | 82.25(+54.75)% | 4.00% | 55.50(+51.50)% |
| Claude-4-Sonnet-20250514-thinking | 36.00% | 91.00(+55.00)% | 3.50% | 67.50(**+64.00**)% |
| DeepSeek-R1 | 39.00% | 88.25(+49.25)% | 3.50% | 60.75(+57.25)% |
| Gemini-2.5-Pro | 59.75% | 96.50(+36.75)% | 11.50% | 53.25(+41.75)% |

We compare the performance on test-generation task and problem-solving task. A submission that passes all official test cases constitutes a successful solution. We report Pass@1,the proportion of problems solved with one attempt, and we show in parentheses **Diff = Pass@1 - VMR**.

From the table, we observe that Pass@1 is much higher than the verdict matching rate, with an average gap of approximately 40%. When a model solves a problem but fails to generate the correct tests for that problem, the model likely does not fully understood the task. The results suggest that LLMs have a relatively serious problem of problem memorization, that is, **they may rely more on memorization than on genuine reasoning or problem understanding**. Pass@1 is lower in C++ than in Python, mirroring the VMR trend. We analyze representative cases in Section 4.1.2.

# 4  ANALYSIS

## 4.1  QUALITATIVE ANALYSIS

### 4.1.1  ANALYSIS OF UNMATCHED CASES

We analyze logs of unmatched cases and group the mismatch causes into three types: (1) Inference Errors, (2) Compilation Errors, and (3) Timeouts. For a detailed analysis with additional examples, see Appendix E.

**(1) Inference Errors**

For non-thinking models, we prompt them to include a chain of thought before generating test code. For thinking models, we extract their reasoning content and analyze them. Our analysis indicates that the model generates only a few incorrect test cases, while most of the remaining cases are correct. However, test code can judge a code submission only when all test cases are correct. Below, we analyze one unmatched case from Doubao-Seed-1.6-thinking-250615.

Figure 3: Sample of unmatched cases

In this unmatched case, let $k$ denote the umbrella price and $r$ the change share. The task reduces to finding the smallest positive integer $m$ such that $(k \cdot m) \bmod 10$ equals 0 or $r$. The model generates 20 test cases, one of which is incorrect. This analysis highlights error and two correct cases (see Figure 3). In case 1 ($k = 25, r = 5$), the smallest valid $m$ is 1, since $25 \times 1 = 2 \times 10 + 5$. In case 2 ($k = 25, r = 3$), the smallest valid $m$ is 2, since $25 \times 2 = 5 \times 10$. In case 3 ($k = 19, r = 7$), the model returned $m = 9$, but the smallest valid is $m = 3$, because $19 \times 3 = 5 \times 10 + 7$.

Due to these inference errors, the system judged the entire code submission incorrect. **Such errors occur when a required condition is overlooked**, such as the minimality constraint above. Despite multiple checks, the model failed to identify its own reasoning flaws.

**(2) Compilation Errors**

Python does not require compilation. In this study, we treat syntax errors as "compilation errors" based on the error messages in the test logs. We observe that **correct reasoning can still produce code with syntax errors**. The common causes are: (i) missing symbols in generated test-case

strings, which lead to unmatched parentheses; (ii) repetitive output that overruns the `max_tokens` limit, leaving code blocks incomplete; and (iii) improper escaping or symbol use—e.g., unescaped double quotes inside double-quoted strings—which truncates the case.

**(3) Timeout**

We set a timeout of 10 seconds for each code execution test. If no test report is returned within this period, the case is marked as a timeout. By analyzing the logs, we find that there are mainly two reasons for timeouts. (i)some submissions contain potential infinite loops, and the LLM-generated test cases trigger them; (ii) some submissions fails to handle empty inputs. When an empty string is passed, the program waits indefinitely for input through functions such as `input` or `cin`, and the case is judged as timeout.

### 4.1.2 ANALYSIS BETWEEN TEST GENERATION AND PROBLEM SOLVING

To further elucidate the phenomenon of problem memorization, we analyze the logs of test generation and problem-solving on the same question. We select a problem of decoding morse code for analysis. For additional analyses, see Appendix E.

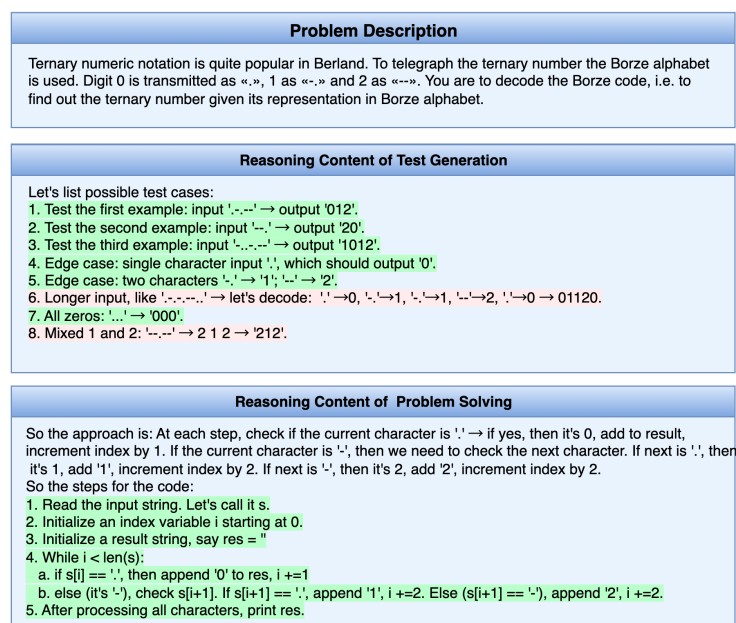

Figure 4: Sample of successfully solving the problem but generating incorrect test code

We analyze logs from Doubao-Seed-1.6-thinking-250615. Figure 4 shows the logs for test generation and problem solving. The problem of decoding morse code can be summarized as mapping strings into numbers, `"."` maps to 0, `"-."` maps to 1, `"--"` maps to 2. The logs reveal two failures. In test case 6, the final `"."` is not decoded, so a character remains and the output does not match the expected `"011200"`. In test case 8, `"-"` is decoded twice, so a 0 is read as 1, and the output does not match the expected `"202"`. However, in problem-solving task, the logic of the code is correct. First, traverse the string character by character, if it is `"."`, directly output 0. If it is `"-"`, continue to judge the next character. Then if the next character is `"-"`, output 1; otherwise, output 2. However, the solution approach is entirely correct.

### 4.2 QUANTITATIVE ANALYSIS

#### 4.2.1 CASE CLASSIFICATION AND PROPORTION OF MATCHED CASES

We require each case to include both passing and failing code submissions. We partition the dataset by the number of failing submissions for each case and compute the proportion of matched cases in

each partition. We selected Genimi-2.5-Pro for visual analysis. Figure 15a reports, for Python, the distribution by number of failing submissions and the corresponding proportion of matched cases; Figure 15b shows the same for C++.

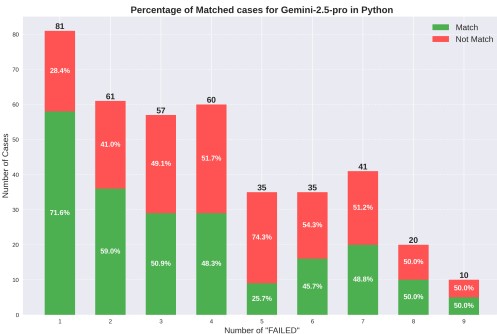

(a) Proportion of Matched cases in Python

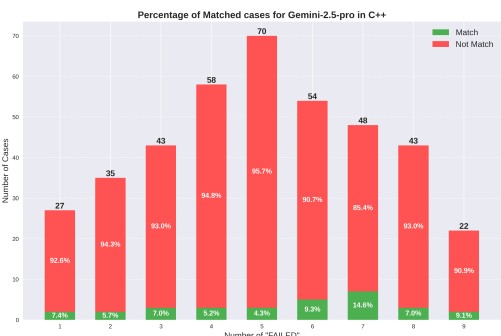

(b) Proportion of Matched cases in C++

Figure 5: Proportion of Matched cases for Gemini-2.5-Pro. Overall, Python datasets have fewer failing code submissions and a higher proportion of matched case than C++.

In Figure 15a, most Python cases contain fewer than five failing code submissions. The dataset shows a triangular-shaped distribution. The proportion of match cases is higher when there are fewer failing code submissions. With one failing submission, the proportion of matched cases achieves 71.6%, higher than in cases with more failing code submissions. This pattern suggests that **more failing code submissions reflect more subtle logic errors, which makes it harder to generate tests that match all submissions**.

In Figure 15b, there are the highest number of cases with five failing code submissions. The distribution peaks in the middle and declines toward both ends. In nearly all cases, the proportion of matched cases is below 15%.

### 4.2.2 EFFECT OF THE NUMBER OF CODE SUBMISSIONS IN EACH CASE

In this section, we analyze the effect of the number of code submissions included in each case on the Verdict Matching Rate. We analyze two models: Doubao-1-5-Pro and Gemini-2.5-Pro (see Figure 6). It shows that the Verdict Matching Rate (VMR) falls as the number of code submissions per case increases, which aligns with expectations since **more submissions make the task harder and lead to a steady decrease in VMR**.

Surprisingly, the largest drop occurs when the number of code submissions rises from one to two. **With a single submission, randomness can produce a spurious match**. Two ambiguous outcomes illustrate this issue: (1) the model predicts "fail" and the code fails, but we cannot

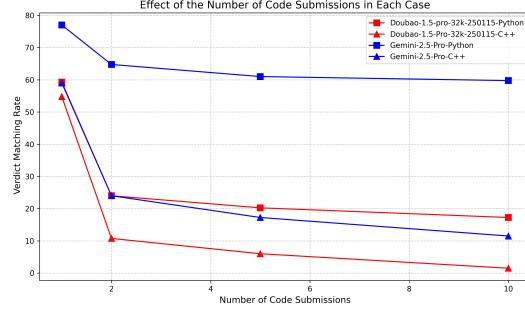

Figure 6: Effect of the Number of code submission in Each Case

tell whether the generated tests are correct or merely flawed; (2) the model predicts "pass" and the code passes, but we cannot tell whether the tests are comprehensive, since reusing input-output examples without new tests may still match. **Requiring both passing and failing submissions in each case significantly reduces this randomness**.

## 5 RELATED WORK

**Test Generation Benchmark for LLMs.** LLMs are widely used for code generation. In order to objectively evaluate the capability of LLMs in code generation, many code generation benchmarks have emerged, such as HumanEval (Chen et al., 2021), MBPP (Austin et al., 2021), and SWE-BENCH (Jimenez et al., 2023).

Code generation in the testing field has received more attention. Extensive research has been conducted in test generation benchmark, including traditional approaches(Cadar et al., 2011; Fraser & Arcuri, 2011) and deep learning-based methods (Dinella et al., 2022; Tufano et al., 2020). As LLMs are adopted for test generation, researchers have proposed numerous benchmarks to evaluate LLM performance on this task. TestBench (Zhang et al., 2024) selected 108 Java programs to evaluate the effect of different context types on test generation. TestEval (Wang et al., 2024) mainly uses coverage as metrics to analyze the test generation capability of different difficulty LeetCode problems. Partial benchmarks have shifted from function level to repository level. TestGenEval (Jain et al., 2024) built an evaluation dataset from a Python repository based on the images of SWE-Bench. It evaluated the performance of the model in test generation and test completion. ProjectTest (Wang et al., 2025) covers the capability of test generation for multiple languages (Python, Java, JavaScript). CLOVER (Xu et al., 2025) focuses mainly on the effect of context length on the capability to generate test code.

Although these benchmarks provide evaluation metrics for test generation, they all overlook the most important evaluation capability, which is the capability to correct subtle logical errors. Although existing benchmarks provide mutation scores, they still focus only on the correction capability of macro code blocks.

## 6 CONCLUSION

We introduce TestJudge, the first benchmark for test generation that targets subtle logical errors. It provides review datasets for Python and C++ with 8,000 code submissions from the Codeforces platform. To reduce evaluation randomness, each case contains ten submissions. We evaluated 13 state-of-the-art models on TestJudge. Although Gemini-2.5-Pro is the top performer, its verdict matching rate reaches only 59.75% on Python and 11.50% on C++, indicating that **LLMs still struggle to generate effective test code**.

To quantify the gap between test generation and problem solving, we compare their success rates on the same set of problems. Problem-solving consistently achieves much higher success rates than test generation, which supports the phenomenon of **problem memorization - models often rely on recalled patterns rather than robust reasoning or task understanding**.

## 7 DISCUSSIONS

**Problem Memorization.** Models might depend on memorizing problems instead of developing robust testing strategies, which underscores a key limitation in existing automated test generation methods. Most models are trained on problem-solving data and recall step-by-step code patterns while neglecting full logical reasoning. By contrast, each test generation records explicit reasoning for each question, so the data are high quality. To reduce memorization, we propose training on two sources: data for problem-solving and test generation. This joint training builds forward and reverse understanding and improves the ability of code generation.

**Function-Level benchmark.** TestJudge uses code from an online programming platform, which differs from many real-world settings. However, we argue that current LLMs perform poorly on function-level test generation, so discussion of repository-level testing is premature. A repository-level dataset could be built from sequences of commits within a single pull request. Although the construction cost is high and requires operations such as pulling, image building, and verification, mature tools exist. Function-level settings remain valuable. The model trained based on the evaluation results can help grade programming problems, provide tests for corpora that lack them, enable sandbox quality checks, and improve the quality and utility of pre-training corpora.

## ETHICS STATEMENT

As large models improve at code generation, more tasks will rely on LLMs. However, test generation is exacting, and even a single faulty test can have serious consequences. LLM-generated tests may label defective code as correct. Therefore, the generated tests require further verification by developers. Due to the potential negative social risks of automated test generation, we hope that practitioners in related fields can recognize both its benefits and its drawbacks.

## REPRODUCIBILITY STATEMENT

In the source code we submitted, we provided all the corresponding code for TestJudge, as well as the evaluation data files. Readers can reproduce the results of our test generation, problem-solving, and direct judgment tasks. We plan to open-source the code and documentation and maintain a public benchmark leaderboard. We will also establish a discussion forum to guide future improvements in automated test generation.

## LLM USAGE STATEMENT

In this study, large language models (LLMs) were used only as auxiliary tools for two tasks: manuscript editing and table formatting. For editing, LLMs improved clarity, corrected grammar, and enhanced readability. For tables, they adjusted structure, standardized notation, and improved layout to aid interpretation. LLMs were not involved in core research activities such as idea development, experimental design, data analysis, interpretation, or scientific writing. All concepts, methods, results, and conclusions are the authors' independent, original work.

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

# APPENDIX

## A  BENCHMARK STATISTICS

### A.1  RULE-BASED FILTERING

We begin with open-source datasets and augment them with data collected by web crawling. We first apply a shell script to remove code submissions with missing fields. We then use the GPT-4.1 and Doubao-1.5-Pro-32K to screen the remaining samples using simple, manually defined rules:

(a) The problem statement is unclear and lacks actionable information;

(b) The problem requires heavy computation that large language models (LLMs) cannot complete;

(c) The problem admits multiple valid solutions, so success cannot be judged by string matching.

### A.2  TAGS OF PROBLEMS

Each problem includes a tag that specifies the required algorithm type. Aggregating these tags provides a quick view of the types of problem and the composition of the evaluation set. Due to the fact that each problem may have multiple tags, the total number of tags will exceed the number of problems. Figure 7 reports the tag statistics for the Python dataset; Figure 8 reports the tag statistics for the C++ dataset.

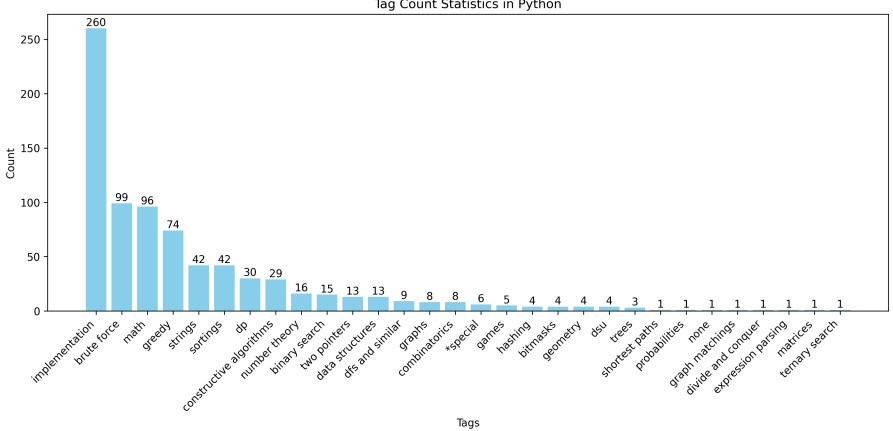

Figure 7: Tag Count Statistics in Python

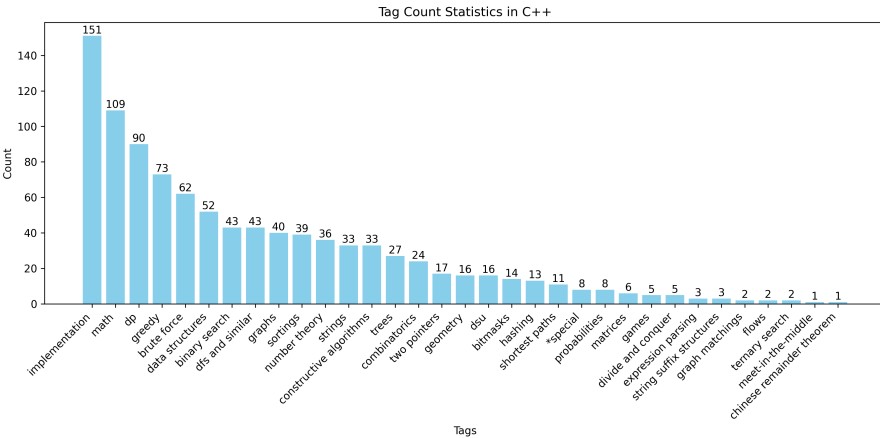

Figure 8: Tag Count Statistics in C++

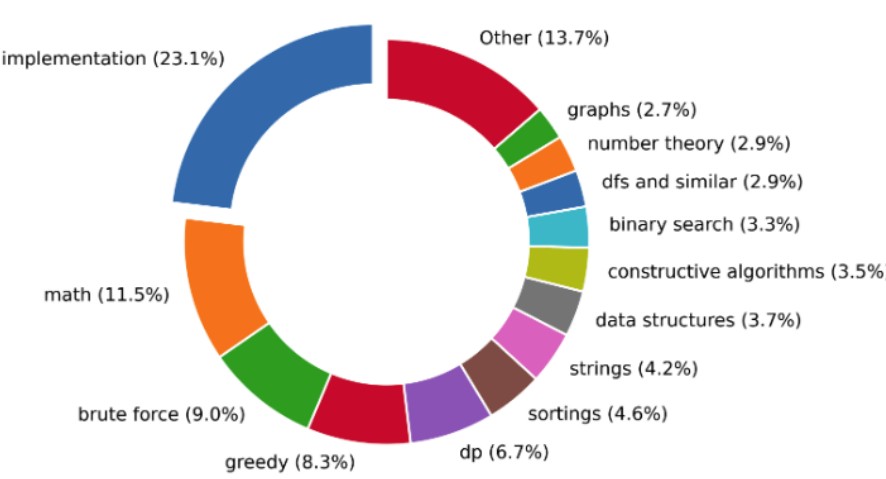

Figure 9: Proportion of TestJudge tags

Figure 9 shows the proportion of each tag among all tags. In the TestJudge, the most common tag is implementation. The second and third most frequent tags are math and brute force. These results suggest that **TestJudge contains many simple problems, and only a few that require complex decomposition and step-by-step reasoning**, such as dynamic programming. Below, we describe the main tag categories.

**(1) Implementation.** Implementation denotes problems that translate logic or algorithms into code and usually do not require complex data structures or algorithm design. These tasks stress precise reading, careful handling of details, and step-by-step execution.

**(2) Math.** Math covers problems that rely on mathematical knowledge, including algebra, elementary number theory, and basic combinatorics. These tasks aim to simplify the problem using formulas, derivations, or known laws rather than ad-hoc programming.

**(3) Brute Force.** Brute force means solving the problem by enumerating all possible cases; it is feasible when the data range is small. Such tasks may not need algorithmic optimization and can be solved by direct search or traversal, but the time complexity must remain feasible.

**(4) Greedy.** Greedy problems use a greedy strategy: choose the locally optimal option at each step to approach a global optimum. They require correct greedy criteria and a justification of why the criteria work.

**(5) Dynamic Programming (Dp).** Dynamic programming solves a problem by dividing it into subproblems and caching their solutions. The key is to define the state and its transition relation. DP is suitable when subproblems overlap and the problem has an optimal substructure.

**(6) Sortings.** Sorting problems center on solving tasks by sorting, either by applying a sorting algorithm directly or by sorting data before further steps. Common methods include quicksort, mergesort, and custom comparators.

**(7) Strings.** String problems focus on string processing, such as concatenation, splitting, matching, replacement, traversal, and character counts, often with hashing, prefix functions, or regular expressions.

## B  PROMPTS

In the test-generation task, we provide problem description, input–output specifications, and input–output samples, ten code submissions with known correctness labels. We then prompt LLMs to

generate test code in a specified format and wrap the output in code blocks for later extraction (see Figure 10).

---

**System prompt**

```
You are an expert in writing test code, and your task is to
generate corresponding test code based on problem information.
```

---

**Prompt for the task of generating test codes in Python experiment**

```
Description of programming problem is:
{problem-description}

These are multiple code submissions.  Each code block is a code
submission.
```python
{code-list}
```
The corresponding evaluation results for the above codes are
{verdict-list}
Among them, "OK" indicates that the code has passed all test cases,
and "FAILED" indicates that there are test cases that did not pass.

The input specification for the code submissions is:
{input-specification}
The output specification for the code submissions is:
{output-specification}
These are multiple sets of example inputs, please design your
input-str based on them:
{demo-input}
These are multiple sets of corresponding example outputs, please
design your output-str based on them:
{demo-output}

Please generate unit test code that can pass all tests on code
submission that meets the requirements of the question, and at
least one sample that does not meet the requirements of the
question.
The generated unit test code is strictly written in the following
format:
```python
{format of generated code}
```

When generating unit test code, strictly follow the following
rules:
1.When generating unit test code, please do not repeatedly
encapsulate into code blocks
2.Please do not write the content of the code submission into the
test function.  Please note that calling run_method will obtain the
output from the input input_str to the code submission.  Please do
not modify the function name, keep it as run_method().
3.Please refer to the examples of input and output for the code
when designing input_str and output_str.
4.Please check the syntax and indentation to ensure that the
program can execute.
```

Figure 10: Prompt for test-generation task.

In the problem-solving task, we provide problem description, input–output specifications, and input–output samples to generate the corresponding solution code (see Figure 11).

---

**System prompt**

```
You are an expert in coding competitions, and your task is to
generate code that can solve problems based on problem information.
```

---

**Prompt for the task of generating test codes in Python experiment**

```
Description of programming problem is:
{problem-description}

The input specification for the code submissions is:
{input-specification}
The output specification for the code submissions is:
{output-specification}

These are multiple sets of example inputs, please design your
input-str based on them:
{demo-input}
These are multiple sets of corresponding example outputs, please
design your output-str based on them:
{demo-output}

Please write code that can solve the problem based on the task
description, input-output specifications, and input-output
examples.
More test cases similar to input-output examples will be used for
verification, and the code you write needs to pass all the cases.
The generated code is strictly written in the following format:
```python
{format of generated code}
```

When generating code that can solve the problem, strictly follow
the following rules:
1.Please read the data from the standard input and write it to the
standard output, using input() and print().
2.Please check the syntax and indentation to ensure that the
program can execute.
```

Figure 11: Prompt for problem-solving task.

We also introduce a task that outputs inference results directly (see Setion D.5) for comparison with the test-generation task. In inference-only task, we provide problem description and code submissions to LLMs. We then prompt them to generate a pass/fail verdict (see Figure 12).

---

**System prompt**

```
You are an expert in code evaluation, and your task is to directly
determine the verdicts of the code submissions.
```

---

**Prompt for the task of generating test codes in Python experiment**

```
Description of programming problem is:
{problem-description}

These are multiple submitted codes, including those that meet the
requirements and those that do not.  Each code block is a submitted
code.
```python
{code-list}
```

Please determine whether the above codes can perfectly solve the
corresponding programming problem.  If it can, return the result as
OK; if not, return FAILED.
Please place the judgment results of all submitted codes at the end
of the response.  You need to output the result of each code line
by line.  The judgment result is strictly written in the form of
the following example:
<result>
OK or FAILED
</result>

When generating the verdicts, strictly follow the following rules:
1.You can first output the reason for the judgment, and then
output the final judgment result.  The judgment result starts with
<result> and ends with </result>
2.Please do not include <result> and </result> outside of the
final judgment result.  Please do not add code and description when
outputting the judgment result.
```

Figure 12: Prompt for inference only task.

## C  EVALUATED MODELS

We evaluated 13 state-of-the-art models. Figure 4 lists their providers and open-source status.

Table 4: Large Language Models Evaluated in TestJudge

| Model Provider | Model Name | Type | Thinking |
|---|---|---|---|
| **OpenAI** | GPT-4.1 (OpenAI, 2025) | Proprietary | × |
| | GPT-4o (OpenAI, 2024) | Proprietary | × |
| **DeepSeek** | DeepSeek-V3 (Liu et al., 2024) | Open Source | × |
| | DeepSeek-R1 (Guo et al., 2025) | Open Source | ✓ |
| **ByteDance** | Doubao-1.5-Pro-32K | Proprietary | × |
| | Doubao-1.5-Thinking-Pro | Proprietary | ✓ |
| | Doubao-Seed-1.6-Thinking (Seed et al., 2025) | Proprietary | ✓ |
| **Anthropic** | Claude-3.7-Sonnet (Anthropic, 2025a) | Proprietary | × |
| | Claude-4-Sonnet (Anthropic, 2025b) | Proprietary | × |
| | Claude-3.7-Sonnet-Thinking | Proprietary | ✓ |
| | Claude-4-Sonnet-Thinking | Proprietary | ✓ |
| **Alibaba** | Qwen-33B-A22B (Yang et al., 2025) | Open Source | × |
| **Google** | Gemini-2.5-Pro (Google DeepMind, 2025) | Proprietary | ✓ |

## D  EXPERIMENT DETAILS

### D.1  EXPERIMENT SETUP

To ensure a fair comparison, we use the same parameters for three tasks—test generation, problem solving, and direct judgment—as shown in Table 5. Because thinking models generate longer chains of thought, we set a higher max_tokens for them than for non-thinking models.

Table 5: Experiment Setup

| Parameter | Value |
|---|---|
| **Temperature** | 0.8 |
| **Top_p** | 0.8 |
| **Max_tokens** | 1024 (Non-Thinking Models) |
| | 16384 (Thinking Models) |

### D.2  METRICS

**Verdict Matching Rate (VMR).** Each sample contains a question and ten code submissions. We apply the test code generated by LLMs to ten code submissions. If the ten verdicts match the ground-truth verdicts, we label the sample as a match (see Figure 13).

We select the proportion of matched samples among all samples as our metric. The formula can be expressed as follows:

$$\text{Verdict Matching Rate} = \frac{\text{Number of match samples}}{\text{Total number of samples}} \times 100\% \qquad (2)$$

**Coverage.** Code line coverage is a software-testing metric that reports the proportion of executable lines exercised by tests. High coverage does not guarantee thorough testing; it only shows that lines ran, not that behavior is correct or that all scenarios were examined. Even so, coverage is a useful first indicator for finding untested code that needs further attention.

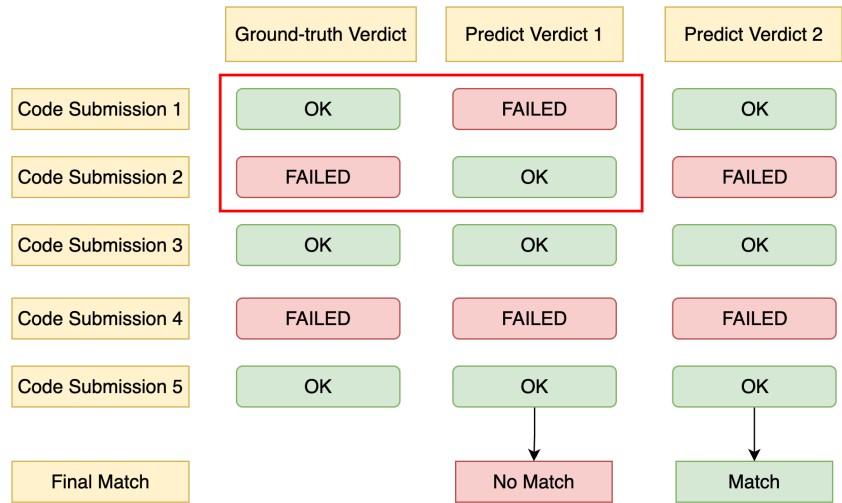

Figure 13: Matching Situation Diagram

**Difference between success rate of problem-solving (Pass@1) and VMR (Diff).** For each problem, we prompt the LLM for a solution and compute Pass@1. Then, we use the difference between Pass@1 and VMR to quantify the severity of problem memorization (see Formula 3).

$$\text{Diff} = \text{Pass@1} - \text{VMR} \tag{3}$$

## D.3 TEST GENERATION

### D.3.1 EVALUATION ON TESTJUDGE

Table 6: Metrics of various LLMs for test generation. The best results are bolded.

| Model | Python VMR | C++ VMR | Python Coverage | C++ Coverage |
|---|---|---|---|---|
| **Non-thinking Models** | | | | |
| Doubao-1.5-Pro-32k-250115 | 17.25% | 1.50% | 97.28% | 90.13% |
| Qwen3-235B-A22B | 15.50% | 2.00% | 97.87% | 90.86% |
| GPT-4o | 11.75% | 0.00% | 97.13% | 90.88% |
| GPT-4.1 | 19.00% | 1.75% | 97.41% | 90.85% |
| DeepSeek-V3 | 12.75% | 1.50% | 97.43% | 90.68% |
| Claude-3.7-Sonnet-20250219 | 19.50% | 1.50% | 97.81% | 91.18% |
| Claude-4-Sonnet-20250514 | 25.25% | 3.75% | 97.69% | 91.24% |
| **Thinking Models** | | | | |
| Doubao-1.5-thinking-Pro-250415 | 47.75% | 2.00% | 97.67% | 91.08% |
| Doubao-Seed-1.6-thinking-250615 | 53.00% | 6.75% | 97.35% | 90.37% |
| Claude-3.7-Sonnet-thinking-20250219 | 27.50% | 4.00% | 97.81% | **91.72%** |
| Claude-4-Sonnet-thinking-20250514 | 36.00% | 3.50% | 97.79% | **91.72%** |
| DeepSeek-R1 | 39.00% | 3.50% | **97.98%** | 90.55% |
| Gemini-2.5-Pro | **59.75%** | **11.50%** | 97.90% | 91.14% |

We evaluated 13 state-of-the-art models on TestJudge. Table 6 reports the Verdict Matching Rate (VMR) and Coverage for each model in the test generation. We found that:

(1) The average VMR of non-thinking models is lower than that of thinking models. The reasoning ability of a thinking model can improve its capacity to generate tests.

(2) Claude-4-Sonnet-20250514 is the best performing non-thinking model, yet its VMR is only 25.25% for Python and 3.75% for C++; Genmini-2.5-Pro is the best performing thinking model, yet its VMR is only 59.75% for Python and 11.50% for C++. 65% of the problems in the Python dataset have implementation tags (which can be solved directly without involving complex reason-

ing), while 37.75% of the problems in the C++ dataset. Genmini-2.5-Pro has a VMR smaller than the proportion of problems labeled with "implementation" in both datasets, suggesting it cannot successfully generate tests for simple questions. This suggests that **LLMs struggle to generate effective test code**.

(3) For the same model, metrics differ by programming language. **VMR and Coverage are much lower in C++ than in Python**, so capability is stronger in Python. This is also consistent with the results of other multilingual test benchmarks. The main reason is that most of the available code data is written in Python. When training models, the proportion of Python data is much higher than that of other languages, leading to weaker code generation capabilities in those languages compared to Python.

(4) From a practical perspective, higher line coverage exercises more execution paths and makes validation more complete, which tends to increase VMR. We construct a scatter plot using VMR and coverage, and fit the scatter points into a straight line. Figure 14 shows **a positive correlation between coverage and VMR**, which is consistent with the actual fact.

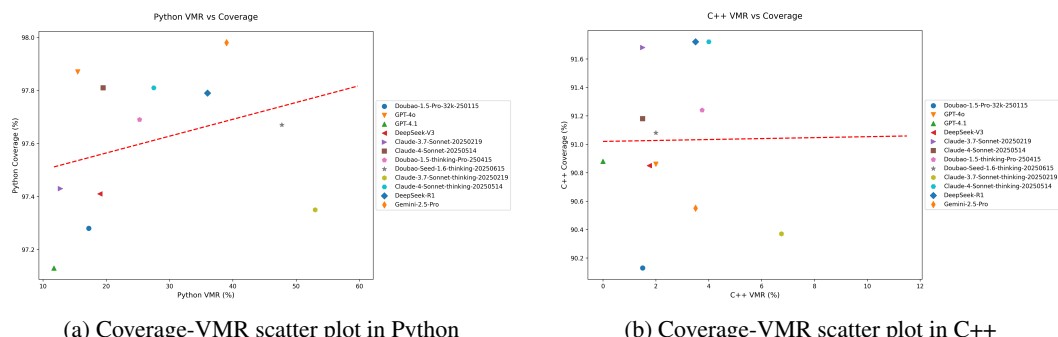

(a) Coverage-VMR scatter plot in Python          (b) Coverage-VMR scatter plot in C++

Figure 14: Coverage-VMR scatter plot. In Python and C++, there is a weak positive correlation between coverage and VMR.

### D.3.2 EVALUATION FOR DIFFERENT NUMBERS OF CODE SUBMISSIONS

We also evaluated various models with different numbers of code submissions.

Table 7: VMR for different numbers of code submissions. The best results are bolded.

| Language | Python VMR | | | | C++ VMR | | | |
| The number of code submissions for each case | 1 code | 2 codes | 5 codes | 10 codes | 1 code | 2 codes | 5 codes | 10 codes |
| --- | --- | --- | --- | --- | --- | --- | --- | --- |
| **Non-Thinking Models** | | | | | | | | |
| Doubao-1.5-Pro-32k-250115 | 59.25% | 24.00% | 20.25% | 17.25% | 54.75% | 10.75% | 6.00% | 1.50% |
| Qwen-235B-A22B | 61.00% | 17.75% | 16.50% | 15.50% | 53.50% | 4.25% | 3.75% | 2.00% |
| GPT-4o | 55.25% | 9.00% | 12.50% | 11.75% | 50.25% | 1.25% | 1.25% | 0.00% |
| GPT-4.1 | 60.50% | 22.25% | 23.25% | 19.00% | 57.25% | 7.25% | 5.75% | 1.75% |
| DeepSeek-V3 | 52.50% | 16.00% | 13.25% | 12.75% | 48.00% | 5.50% | 2.25% | 1.50% |
| Claude-3.7-Sonnet-20250219 | 59.25% | 20.50% | 19.75% | 19.50% | 52.75% | 3.75% | 3.25% | 1.50% |
| claude-4-Sonnet-20250514 | 60.75% | 24.75% | 26.00% | 25.25% | 54.75% | 6.75% | 4.75% | 3.75% |
| **Thinking Models** | | | | | | | | |
| Doubao-1.5-Thinking-Pro-250415 | **77.00%** | **70.25%** | 56.25% | 47.75% | 54.75% | 10.75% | 4.00% | 2.00% |
| Doubao-Seed-1.6-Thinking-250615 | **77.00%** | 62.00% | 56.50% | 53.00% | **59.50%** | 18.25% | 11.00% | 6.75% |
| Claude-3.7-Sonnet-Thinking-20250219 | 65.50% | 40.25% | 37.50% | 27.50% | 52.25% | 8.50% | 6.75% | 4.00% |
| Claude-4-Sonnet-Thinking-20250514 | 64.00% | 43.75% | 41.50% | 36.00% | 55.50% | 12.50% | 7.75% | 3.50% |
| DeepSeek-R1 | 69.00% | 52.25% | 45.50% | 39.00% | 52.00% | 23.75% | 13.25% | 3.50% |
| Gemini-2.5-Pro | **77.00%** | 64.75% | **61.00%** | **59.75%** | 59.00% | **24.00%** | **17.25%** | **11.50%** |

Table 7 shows that the Verdict Matching Rate (VMR) falls as the number of code submissions per case increases, which aligns with expectations since more submissions make the task harder and lead to a steady decrease in VMR.

Surprisingly, the largest drop occurs when the number of code submissions rises from one to two. **With a single submission, randomness can produce a spurious match. Requiring both passing**

**and failing submissions in each case significantly reduces this randomness**. Two ambiguous outcomes illustrate this issue:

(1) the model predicts "fail" and the code fails, but we cannot tell whether the generated tests are correct or merely flawed;

(2) the model predicts "pass" and the code passes, but we cannot tell whether the tests are comprehensive, since reusing input-output examples without new tests may still match.

### D.3.3 EVALUATION FOR DIFFERENT INFORMATION

To assess whether providing code submissions interferes with model test generation, we design three comparison experiments:

Group 1: Only provide problem information;

Group 2: Provide problem information and code submissions without known correctness labels;

Group 3: Provide problem information and code submissions with known correctness labels.

Table 8: VMR for different information.

| Language | Python | | | C++ | | |
|---|---|---|---|---|---|---|
| Experiment Group | Group 1 | Group 2 | Group 3 | Group 1 | Group 2 | Group 3 |
| **Non-Thinking Models** | | | | | | |
| Doubao-1.5-Pro-32k-250115 | 12.50% | 18.25% | 17.25% | 1.00% | 3.50% | 1.50% |
| Qwen-235B-A22B | 10.50% | 15.25% | 15.50% | 1.25% | 1.00% | 2.00% |
| GPT-4o | 8.50% | 8.50% | 11.75% | 0.75% | 0.00% | 0.00% |
| GPT-4.1 | 12.50% | 22.25% | 19.00% | 1.75% | 2.75% | 1.75% |
| DeepSeek-V3 | 5.75% | 12.25% | 12.75% | 0.50% | 1.50% | 1.50% |
| Claude-3.7-Sonnet-20250219 | 11.25% | 17.75% | 19.50% | 0.50% | 1.75% | 1.50% |
| Claude-Sonnet-4-20250514 | 17.50% | 23.50% | 25.25% | 1.50% | 2.50% | 3.75% |
| **Thinking Models** | | | | | | |
| Doubao-1.5-Thinking-Pro-250415 | 45.25% | 50.50% | 47.75% | 3.50% | 2.75% | 2.00% |
| Doubao-Seed-1.6-Thinking-250615 | 40.00% | 46.25% | 53.00% | 5.00% | 5.00% | 6.75% |
| Claude-3.7-Sonnet-Thinking-20250219 | 21.75% | 34.75% | 27.50% | 2.25% | 3.25% | 4.00% |
| Claude-Sonnet-4-Thinking-20250514 | 32.25% | 32.25% | 36.00% | 3.50% | 4.25% | 3.50% |
| DeepSeek-R1 | 22.75% | 40.25% | 39.00% | 3.00% | 4.75% | 3.50% |
| Gemini-2.5-Pro | 45.50% | 53.00% | 59.75% | 6.50% | 7.50% | 11.50% |

In Figure 8, we observe the following:

(1) The VMR of the experimental group without code submission (Group 1) is lower than that of the group with code submission (Group 2,3), code submission can enhance the model's ability to generate tests. Code submission provides error information, allowing the model to generate test cases with broader coverage.

(2) The VMR of the group with code submission and its correctness labels (Group 3) is generally higher than that of the group with code submission alone (Group 2). When the code submission status is unknown, the model may make errors, overlooking certain error types and failing to generate effective test samples.

### D.4 TEST GENERATION VS PROBLEM SOLVING

We compare the performance on test generation and problem solving. A submission that passes all official test cases counts as a successful solution. We report Pass@1, the proportion of problems solved with one attempt, and we show in parentheses **Diff = Pass@1 - VMR**.

From the table, it can be seen that Pass@1 is much higher than the Verdict Matching Rate, with an average gap of about 40%. When a model solves a problem but fails to produce correct test code for that problem, the model likely has not fully understood the task. The results can infer that LLMs have a relatively serious problem of problem memorization, that is, **they may rely more on**

Table 9: Comparison of test generation and problem-solving. The difference between Pass@1 and VMR is in parentheses. The biggest difference is bolded.

| Model | Python VMR | Python Pass@1 | C++ VMR | C++ Pass@1 |
|---|---|---|---|---|
| **Non-thinking Models** | | | | |
| Doubao-1.5-Pro-32k-250115 | 17.25% | 80.25(+63.00)% | 1.50% | 43.25(+41.75)% |
| Qwen3-235B-A22B | 15.50% | 81.25(+65.75)% | 2.00% | 52.75(+50.75)% |
| GPT-4o | 11.75% | 61.00(+49.25)% | 0.00% | 42.00(+42.00)% |
| GPT-4.1 | 19.00% | 82.25(+63.25)% | 1.75% | 47.75(+46.00)% |
| DeepSeek-V3 | 12.75% | 90.25(**+77.50**)% | 1.50% | 53.50(+52.00)% |
| Claude-3.7-Sonnet-20250219 | 19.50% | 74.25(+54.75)% | 1.50% | 49.50(+48.00)% |
| Claude-4-Sonnet-20250514 | 25.25% | 84.50(+59.25)% | 3.75% | 61.50(+57.75)% |
| **Thinking Models** | | | | |
| Doubao-1.5-thinking-Pro-250415 | 47.75% | 85.75(+38.00)% | 2.00% | 62.75(+60.75)% |
| Doubao-Seed-1.6-thinking-250615 | 53.00% | 84.00(+31.00)% | 6.75% | 51.75(+45.00)% |
| Claude-3.7-Sonnet-20250219-thinking | 27.50% | 82.25(+54.75)% | 4.00% | 55.50(+51.50)% |
| Claude-4-Sonnet-20250514-thinking | 36.00% | 91.00(+55.00)% | 3.50% | 67.50(**+64.00**)% |
| DeepSeek-R1 | 39.00% | 88.25(+49.25)% | 3.50% | 60.75(+57.25)% |
| Gemini-2.5-Pro | 59.75% | 96.50(+36.75)% | 11.50% | 53.25(+41.75)% |

**memorization than on genuine reasoning or problem understanding**. Pass@1 is lower for C++ than for Python, mirroring the VMR trend.

The model shows a clear link between test-generation and problem-solving metrics. We construct a scatter plot of the two metrics and fit a linear model. Figure 15 shows a strong positive correlation between Pass@1 and VMR. This suggests that training on test-generation data may improve problem-solving performance, underscoring the need to build such datasets.

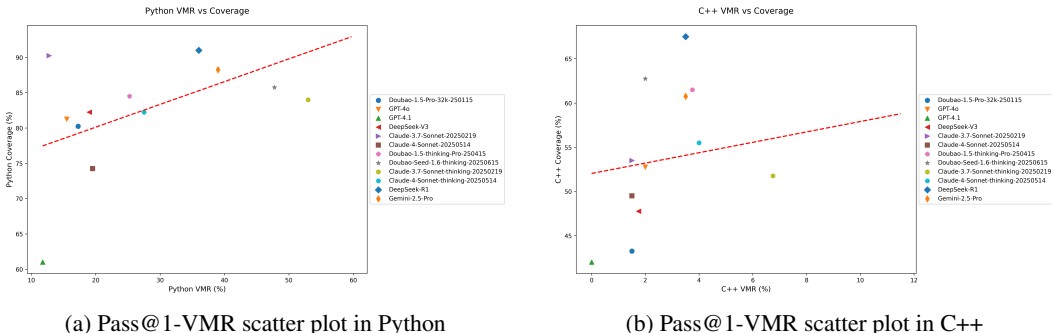

(a) Pass@1-VMR scatter plot in Python  (b) Pass@1-VMR scatter plot in C++

Figure 15: Pass@1-VMR scatter plot. In Python and C++, there is a strong positive correlation between Pass@1 and VMR.

## D.5 TEST GENERATION VS DIRECT JUDGMENT

We prompt the model to directly output the verdicts of the code submissions and calculate the verdict matching rate. We then compare it to the verdict matching rate of test generation.

Table 10: Comparison of Test Generation and Direct Judgment. The difference between the two tasks is in parentheses.

| Number of Code Submissions per Case | 2 Codes | | 5 Codes | | 10 Codes | |
|---|---|---|---|---|---|---|
| Experiment Type | Test Generation | Direct Judgment | Test Generation | Direct Judgment | Test Generation | Direct Judgment |
| Doubao-1.5-Pro-32k-250115 | 23.25% | 65.25% (+42.00%) | 22.75% | 30.00% (+7.25%) | 17.25% | 15.25% (-2.00%) |
| Qwen-235B-A22B | 18.00% | 55.50% (+37.50%) | 16.25% | 25.25% (+9.00%) | 15.50% | 9.75% (-5.75%) |
| GPT-4o | 13.00% | 56.50% (+43.50%) | 12.50% | 25.75% (+13.25%) | 11.75% | 9.25% (-2.50%) |
| GPT-4.1 | 23.75% | 68.75% (+45.00%) | 22.00% | 31.25% (+9.25%) | 19.00% | 14.25% (-4.75%) |
| DeepSeek-V3 | 15.50% | 75.00% (+59.50%) | 13.50% | 29.50% (+16.00%) | 12.75% | 14.50% (+1.75%) |
| Claude-3.7-Sonnet-20250219 | 16.75% | 55.25% (+38.50%) | 19.50% | 17.25% (-1.75%) | 19.50% | 12.75% (-6.75%) |
| Claude-4-Sonnet-20250514 | 23.00% | 59.00% (+36.00%) | 25.75% | 34.50% (+8.75%) | 25.25% | 16.75% (-8.50%) |
| Doubao-1.5-Thinking-Pro-250415 | 68.50% | 76.00% (+7.50%) | 56.00% | 57.50% (+1.50%) | 47.75% | 57.00% (+9.25%) |

Figure 10 shows the difference in VMR between test-generation tasks and direct judgment tasks. As the number of code submissions per sample increases, the VMR for both tasks decreases.

Interestingly, when the sample contains two or five code submissions, the verdict matching rate of direct judgment is generally higher than that of the test-generation task. However, when the sample contains 10 code submissions, the verdict matching rate of the test generation is mostly higher than that of direct judgment, indicating that using generated tests to judge the verdicts of the code submissions is more robust. The reasons for this phenomenon are discussed in Section E.3.

# E  ANALYSIS

## E.1  ANALYSIS OF UNMATCHED CASES

We analyze logs of unmatched cases and group the mismatch causes into three types: (1) Inference Errors, (2) Timeouts, and (3) Compilation Errors.

**(1) Inference Errors**

For non-thinking models, we prompt them to include a chain of thought before generating test code. For thinking models, we extract their reasoning content and analyze it. Our analysis shows the model only generates a few incorrect test cases, while most of the remaining cases are correct. Below, we will present multiple examples of unsuccessful matches due to inference errors.

Table 11 shows the problem of Vasya and String. The task involves modifying characters no more than k times to obtain the max length of consecutive identical substrings.

There are two examples of errors in this task: In case 6, LLMs mistakenly use more than k character transformations, changing 'aaabb' to 'aaaaa', leading to an incorrect maximum length of 5, while the correct answer is 4; in case 10, LLMs again use more than k character transformations, leading to an incorrect maximum length of 6, while the correct answer is 5.

Table 12 shows the problem of Triangle. The task reduces to evaluating four matches: first, whether a triangle can be formed; second, whether a degenerate triangle can be formed. If neither is possible, the output is "IMPOSSIBLE".

There are many possible match pairings, some code submissions overlooked certain cases. A comprehensive test suite should cover all pairings among the four match types. However, the model did not enumerate all pairings, so some erroneous submissions were not judged as FAILED.

**(2) Timeout**

We set a timeout of 10 seconds for each code execution test. If no test report is returned within this period, the case is marked as a timeout. By analyzing the logs, we find that there are mainly two reasons for timeouts. (i)some submissions contain potential infinite loops, and the generated test cases trigger them; (ii) some code fails to handle empty inputs. When an empty string is passed, the program waits indefinitely for input through functions such as `input` or `cin`, and the case times out.

**(3) Compilation Errors of Generated Code**

Python does not require compilation. In this study, we treat syntax errors as "compilation errors" based on the error messages in the test logs. We observe that **correct reasoning can still produce code with syntax errors**. The common causes are: (i) missing symbols in generated test-case strings, which lead to unmatched parentheses; (ii) repetitive output that overruns the `max_tokens` limit, leaving code blocks incomplete; and (iii) improper escaping or symbol use—e.g., unescaped double quotes inside double-quoted strings—which truncates the case.

Table 11: Inference errors result in all code submissions being judged as FAILED

**Case 1: Vasya and String**

**Problem:**
High school student Vasya got a string of length *n* as a birthday present. This string consists of letters 'a' and 'b' only. Vasya denotes beauty of the string as the maximum length of a substring (consecutive subsequence) consisting of equal letters. Vasya can change no more than *k* characters of the original string. What is the maximum beauty of the string he can achieve?

**Input Specification**
The first line of the input contains two integers *n* and *k* ($1 <= *n* <= 100000, 0 <= *k* <= *n*$)— the length of the string and the maximum number of characters to change.The second line contains the string, consisting of letters 'a' and 'b' only.
**Output Specification**
Print the only integer— the maximum beauty of the string Vasya can achieve by changing no more than *k* characters.

**Reasoning Content from Doubao-1.6-Seed-Thinking**
Got it, let's tackle this problem. The user wants unit tests for the submitted codes, where the tests should pass the correct codes (marked OK) and fail the incorrect ones (marked FAILED). First, I need to understand the problem clearly. The problem is about finding the maximum length of a substring of equal characters (either 'a's or 'b's) after changing at most k characters. So the correct approach should handle both cases: turning non-'a's into 'a's (up to k times) to get a long 'a' substring, and similarly for 'b's, then taking the maximum of the two.

Let's think about different test scenarios:
1. **All characters the same**: Input like '5 0\naaaaa' → Output '5'. Correct codes should return 5. Incorrect ones might mess up.
2. **k equals n**: Input '3 3\n aba' → Output '3' (change all to 'a' or 'b'). Correct codes should handle this.
3. **k=0, mixed characters**: Input '5 0\n ababa' → Output '1' (can't change anything, so maximum is 1).
4. **Need to change some characters**: Like the example 1: '4 2\n abba' → 4.
5. **Longer substring with one change**: Example 2: '8 1\n aabaabaa' → 5.
6. **Changing to 'b's instead of 'a's**: Input '5 1\n aaabb' → Changing one 'a' to 'b' gives 'aabbb' (length 3) or changing one 'b' to 'a' gives 'aaaaa' (length 5). So output is 5. Correct codes should check both cases.
7. **Edge case: n=1**: Input '1 0\n a' → Output '1'; '1 1\n b' → Output '1'.
8. **Edge case: k=0, all different**: Input '3 0\n aba' → Output '1'.
9. **Edge case: k=0, consecutive characters**: Input '5 0\n aaabb' → Output '3' (the first three 'a's).
10. **Case where changing to 'b's is better**: Input '6 2\n ababab' → Changing two 'a's to 'b's gives 'bbbbbb' (length 6) or two 'b's to 'a's gives 'aaaaaa' (length 6). So output 6.
11. **Case where you need to slide the window**: For example, '7 2\n abbaabb' → Let's see: the string is 'a b b a a b b'. Changing two 'a's to 'b's: the 'a's are at 0,3,4. Changing 0 and 3 gives 'bbbbabb' (length 4), but changing 3 and 4 gives 'abbbbbb' (length 6). So output 6.

Table 12: Insufficient comprehensive inference resulted in some erroneous code submissions being judged as OK

**Case 1: Triangle**

**Problem:**
Johnny has a younger sister Anne, who is very clever and smart. As she came home from the kindergarten, she told his brother about the task that her kindergartener asked her to solve. The task was just to construct a triangle out of four sticks of different colours. Naturally, one of the sticks is extra. It is not allowed to break the sticks or use their partial length. Anne has perfectly solved this task, now she is asking Johnny to do the same.
The boy answered that he would cope with it without any difficulty. However, after a while he found out that different tricky things can occur. It can happen that it is impossible to construct a triangle of a positive area, but it is possible to construct a degenerate triangle. It can be so, that it is impossible to construct a degenerate triangle even. As Johnny is very lazy, he does not want to consider such a big amount of cases, he asks you to help him.

**Input Specification**
The first line of the input contains four space-separated positive integer numbers not exceeding 100 — lengthes of the sticks.
**Output Specification**
Output TRIANGLE if it is possible to construct a non-degenerate triangle. Output SEGMENT if the first case cannot take place and it is possible to construct a degenerate triangle. Output IMPOSSIBLE if it is impossible to construct any triangle. Remember that you are to use three sticks. It is not allowed to break the sticks or use their partial length.

**Reasoning Content from Doubao-1.6-Seed-Thinking**
Got it, let's tackle this problem. The user wants unit tests for the submitted codes that check if they meet the problem requirements. First, I need to understand the problem thoroughly. The problem is about four sticks, and we need to find if any three can form a non-degenerate triangle (TRIANGLE), if not then a degenerate one (SEGMENT), else IMPOSSIBLE. The key points are:
- For a non-degenerate triangle, the sum of any two sides must be greater than the third. But since we can sort, the main condition is that the sum of the two smaller sides is greater than the largest.
- For a degenerate triangle, the sum of the two smaller sides equals the largest.
- We need to check all combinations of three sticks from four.

Now, let's generate the unit tests. The unit test code should have test methods for each of these cases. The test methods should use input_str and output_str as per the problem's input and output specifications.

For example:
- Test case 1: Input '4 2 1 3' → Output 'TRIANGLE'
- Test case 2: Input '7 2 2 4' → Output 'SEGMENT'
- Test case 3: Input '3 5 9 1' → Output 'IMPOSSIBLE'
- Test case 4: Input '1 2 3 4' → Output 'TRIANGLE'
- Test case 5: Input '2 3 4 5' → Output 'TRIANGLE'
- Test case 6: Input '1 2 3 5' → Output 'SEGMENT'
- Test case 7: Input '1 2 3 6' → Output 'SEGMENT'
- Test case 8: Input '1 3 4 5' → Output 'TRIANGLE'
- Test case 9: Input '1 2 3 3' → Output 'TRIANGLE'
- Test case 10: Input '1 2 4 7' → Output 'IMPOSSIBLE'
- Test case 11: Input '1 1 2 3' → Output 'SEGMENT'

These test cases cover various scenarios: non-degenerate in different triplets, degenerate in different triplets, no non-degenerate/no degenerate, multiple non-degenerate, multiple degenerate.

## E.2   ANALYSIS BETWEEN TEST GENERATION AND PROBLEM SOLVING

To better understand the phenomenon of problem memorization, we examine the logs of test generation and problem-solving for the same question.

Figure 13 shows the logs for test generation and problem solving. The problem of morse code decoding can be summarized as converting strings into numbers, `"."` is 0, `"-."` is 1, `"--"` is 2.

The logs reveal two failures. In case 6, the final `"."` is not decoded, so a character remains and the output does not match the expected `"011200".`; In case 8, `"-"` is decoded twice, so a 0 is read as 1 and the output does not match the expected `"202"`. But when solving problems, the logic of the code is correct. First, traverse the string line by line, if it is `"."`, directly output 0. If it is `"-"`, continue to judge the next character. Then if the next character is `"-"`, output 1, otherwise, output 2.

However, the solution approach is entirely correct. We iterate through the string with an index. At each step, check if the current character is '.' $\rightarrow$ if yes, then it's 0, add to result, increment index by 1. If the current character is '-', then we need to check the next character. If next is '.', then it's 1, add '1', increment index by 2. If next is '-', then it's 2, add '2', increment index by 2.

Figure 14 shows the New Year Transportation problem. Each cell stores a value that tells how many steps to move forward. Starting from cell 1, we advance step by step and check whether we reach the target cell.

The logs reveal two failures. In case 5, moving from cell 1 to 3 and then to 4 reaches the target, so the output should be YES, not NO; in case 11, moving from cell 1 to 4 and then to 5 also reaches the target, so the output should be YES, not NO.

However, the solution approach is entirely correct. We simulate the journey starting from cell 1. We keep moving forward by following the one - way portals. In each step, we determine the next cell based on the current cell's portal value. The loop runs as long as the current position is less than the target cell t. Once the loop ends (when the current position is greater than or equal to t), we check if the current position exactly matches t to determine if we can reach the target.

Table 13: Generate error tests but can solve the problem

**Case 1: Borze**

**Problem:**
Ternary numeric notation is quite popular in Berland. To telegraph the ternary number the Borze alphabet is used. Digit 0 is transmitted as '.', 1 as '-.' and 2 as '–'. You are to decode the Borze code, i.e. to find out the ternary number given its representation in Borze alphabet.

**Input Specification**
The first line contains a number in Borze code. The length of the string is between 1 and 200 characters. It's guaranteed that the given string is a valid Borze code of some ternary number (this number can have leading zeroes).
**Output Specification**
Output the decoded ternary number. It can have leading zeroes.

**Reasoning Content from Doubao-1.6-Seed-Thinking for test generation**
Okay, I need to generate unit test code for the Borze code decoding problem. Let's start by understanding the problem. The task is to decode a Borze code string into a ternary number. The Borze code uses: '.' for 0; '-.' for 1; '–' for 2.
So first, I need to create test cases based on the example inputs and outputs, and possibly other edge cases. Let's list possible test cases:
1. Test the first example: input '.-.–' → output '012'.
2. Test the second example: input '–.' → output '20'.
3. Test the third example: input '-..-.–' → output '1012'.
4. Edge case: single character input '.', which should output '0'.
5. Edge case: two characters '-.' → '1'; '–' → '2'.
6. Longer input, like '.-.-.–..' → let's decode:'.' →0, '-.'→1, '-.'→1, '–'→2, '.'→0 → '01120.'
7. All zeros: '...' → '000'.
8. Mixed 1 and 2: '-.-.-.' → 1 1 1 → '111'; '–.–' → 2 1 2 → '212'.

**Reasoning Content from Doubao-1.6-Seed-Thinking for problem solving**
Okay, let's see. The problem is to decode Borze code into a ternary number. The Borze alphabet uses specific symbols for 0, 1, 2. So first, I need to remember what each code represents. Oh right, 0 is '.', 1 is '-.', and 2 is '–'.
The input is a string of Borze code, and I have to convert that into the corresponding ternary digits. The output is the decoded number as a string, which can have leading zeros.

So the approach is: iterate through the string with an index. At each step, check if the current character is '.' → if yes, then it's 0, add to result, increment index by 1. If the current character is '-', then we need to check the next character. If next is '.', then it's 1, add '1', increment index by 2. If next is '-', then it's 2, add '2', increment index by 2.
So the steps for the code:
1. Read the input string. Let's call it s.
2. Initialize an index variable i starting at 0.
3. Initialize a result string, say res = ''
4. While i < len(s):
a. if s[i] == '.', then append '0' to res, i +=1
b. else (it's '-'), check s[i+1]. If s[i+1] == '.', append '1', i +=2. Else (s[i+1] == '-'), append '2', i +=2.
5. After processing all characters, print res.

Table 14: Generate error tests but can solve the problem (Continue).

**Case 2: New Year Transportation**

**Problem:**
New Year is coming in Line World! In this world, there are *n* cells numbered by integers from 1 to *n*, as a $1<=$*n* board. People live in cells. However, it was hard to move between distinct cells, because of the difficulty of escaping the cell. People wanted to meet people who live in other cells.
So, user has made a transportation system to move between these cells, to celebrate the New Year. First, he thought of *n*-1 positive integers. For every integer *i* where $1<=$*i*$<=$*n*-1 the condition holds. Next, he made *n*-1 portals, numbered by integers from 1 to *n*-1. The *i*-th $(1<=$*i*$<=$*n*-1) portal connects cell *i* and cell (*i*+$a^i$*), and one can travel from cell *i* to cell (*i*+$a^i$*) using the *i*-th portal. Unfortunately, one cannot use the portal backwards, which means one cannot move from cell (*i*+$a^i$*) to cell *i* using the *i*-th portal.
Currently, I am standing at cell 1, and I want to go to cell *t*. However, I don't know whether it is possible to go there. Please determine whether I can go to cell *t* by only using the construted transportation system.

**Input Specification**
The first line contains two space-separated integers *n* ($3<=$*n*¡=$310^4$) and *t* ($2<=$*t*$<=$*n*) — the number of cells, and the index of the cell which I want to go to. The second line contains *n*-1 space-separated integers. It is guaranteed, that using the given transportation system, one cannot leave the Line World.
**Output Specification**
If I can go to cell *t* using the transportation system, print YES

**Reasoning Content from Doubao-1.6-Seed-Thinking**
So, to summarize, the unit tests should include:
1. Exact reach in one step: Input: '5 3\n2 1 1 1'. Output: YES.
2. Exact reach in two steps: Input: '6 5\n1 3 1 1 1'. Output: YES.
3. Exact reach in multiple steps: Input: '8 4\n1 2 1 2 1 2 1'. Output: YES.
4. Overshoot in one step: Input: '5 3\n3 1 1 1'. Output: NO.
5. Overshoot in two steps: Input: '6 4\n2 2 1 1 1'. Output: NO.
6. Reach t=n in one step: Input: '10 10\n9 1 1 1 1 1 1 1 1'. Output: YES.
7. Reach t=n in multiple steps: Input: '7 7\n1 1 1 1 1 1'. Output: YES.
8. Overshoot and can't reach t: Input: '8 5\n1 2 1 2 1 1 1'. Output: NO.
9. Path ends at t=n: Input: '9 9\n3 1 2 1 1 1 1 1'. Output: YES.
10. T is in the middle of the path: Input: '10 6\n2 1 2 1 1 1 1 1 1'. Output: YES.
11. T is not in the path and not overshot: Input: '7 5\n3 1 1 1 1 1'. Output: NO.
12. Edge case: Input: '5 2\n1 1 1 1'. Output: YES.
13. Edge case: Input: '5 2\n2 1 1 1'. Output: NO.

**Reasoning Content from Doubao-1.6-Seed-Thinking for problem solving**
Here is the step-by-step logic:
1. Initialize the current position to cell 1.
2. Enter a loop that continues as long as the current position is less than the target cell t.
3. Inside the loop, determine the next cell by adding the portal's value a[i] to the current cell i. Since the cell numbers are 1-indexed and Python lists are 0-indexed, the jump from current_cell is a[current_cell-1].
4. Update the current position to this new cell.
5. The loop terminates when the current position is either equal to or greater than t.
6. After the loop, if the current position is exactly t, it means we have successfully reached the destination. We print "YES".
7. If the current position is greater than t, it means we have overshot the target. Since all movements are forward, we can never go back to t. In this case, we print "NO".

### E.3 ANALYSIS BETWEEN TEST GENERATION AND DIRECT JUDGMENT

Table 15: Analysis of the direct judgment task.

**Reasoning Content from Doubao-1.6-Seed-Thinking**
Let's analyze each code one by one:
1. First code - OK:
- Correctly counts the frequency of each bar length and finds the maximum frequency (tower height) and number of unique lengths (number of towers).
2. Second code - FAILED:
- The logic is overly complicated and incorrect. It tries to count consecutive same elements after sorting, but the implementation is wrong.
3. Third code - FAILED:
- The logic is completely wrong and doesn't solve the problem. It's doing unnecessary string manipulations and incorrect counting.
4. Fourth code - OK:
- Correctly finds the maximum count and number of unique elements.
5. Fifth code - FAILED:
- The logic is incorrect. It tries to count unique elements and maximum frequency but does it in a wrong way (count is incremented incorrectly).
6. Sixth code - OK:
- Correctly counts frequencies using a fixed-size array and finds the maximum count and number of unique elements.
7. Seventh code - OK:
- Similar to sixth code but uses 1001 as fixed size. Works correctly for the problem constraints.
8. Eighth code - OK:
- Correctly uses set to count unique elements and finds maximum frequency.
9. Ninth code - FAILED:
- The logic is partially correct but the height counting is wrong (only increments by 1 for duplicates).
10. Tenth code - FAILED:
- The logic is incorrect. The formula (n-len(a))+1 doesn't correctly calculate the maximum height.

Figure 15 shows that the **model evaluates submissions by aligning them with the problem's solution logic rather than generating individual tests**. Direct judgment task depends heavily on the execution logic extracted from the code submissions. When many submissions are present, extracting incorrect problem logic from any one of them can cause the sample to fail. **Generating tests to determine verdicts is not subject to this randomness**. This also explains that the metrics of the test-generation task surpass those of the direct-judgement task as the number of code submissions increases.

