# OpenReview forum: "TestJudge: A Rigorous Benchmark for Unit Test Generation and Quality Assessment"
_ICLR.cc/2026/Conference — ICLR 2026 Conference Withdrawn Submission_

### Official Review · Reviewer_vnLy · 2025-10-28

**Soundness:** 1
**Presentation:** 1
**Contribution:** 2
**Rating:** 2
**Confidence:** 5

**Summary:**

This paper presents TestJudge, a unit test–generation benchmark that evaluates test correctness, usefulness, and bug detection. Test correctness in TestJudge is measured by the VMR of the generated unit tests, usefulness is measured by line coverage, and bug detection is claimed but not justified or presented. Each problem in TestJudge is a triple containing three components: a problem description, an input–output specification (description), and input–output samples. The benchmark contains two tasks: test generation and problem solving.

Test generation prompts the LLMs with the problem triple and 10 correct code submissions, and the LLMs generate a test suite (i.e., multiple test cases). Problem solving prompts the LLMs with the problem triple and has the LLMs generate solutions. The authors present a new metric for test generation, Verdict Matching Rate (VMR), which measures the percentage of all 10 correct code submissions that pass the LLM-generated unit tests. Test generation is measured using VMR and line coverage. Problem solving is measured using Pass@1. The authors further develop a new metric, Diff = Pass@1 − VMR, to measure whether LLMs rely on memorization rather than genuine reasoning or problem understanding. The authors evaluated 13 LLMs, both reasoning and non-reasoning, on the benchmark and provide qualitative analysis of the results.

##

**Strengths:**

1. TestJudge is a large dataset, containing 800 problems and two different mainstream languages, Python and C++.

2. Quantitative analysis of the results offers some insight into generating tests for submissions with logic errors.

**Weaknesses:**

1. Why do the authors choose to provide all 10 code submissions in the prompt to the LLM? This choice is not well justified. Function-level test generation usually only utilizes the description, documentation, and the function under test as the prompt. Having multiple code submissions seems redundant and might lead to context rot.

2. VMR could be explained in simpler terms: instead of one function under test passing one generated test suite, VMR measures whether all 10 semantically identical code submissions pass the same generated test suite. Recalling W1, one choice is to simply sample 1 + 10 = 11 correct code submissions, let the LLM generate tests on the first, and evaluate on the other 10 submissions.
3. The logic correction stated in Section 2.4 is never justified nor presented with result statistics in Section 3. To make sense of it, the authors could use pairs of correct and incorrect submissions. The task, in the setting of TestJudge, could let the LLMs generate a test suite, verify the test’s correctness using VMR, and then run the incorrect submission to see if it fails.
4. The Diff metric is not convincing. Generating solutions and generating tests are fundamentally different, so measuring their difference can only reveal the difference in task difficulty, not memorization. If the authors aim for this direction, they should consider measuring the performance difference of the *same* task before and after some perturbations, for example in mathematical reasoning [1] and program semantics reasoning [2].
5. The overly high coverage is suspect. I think it is caused by the nature of coding contest problems, where each solution is short. In Figure 1, both submissions are very short, less than 5 lines. In this case, line coverage, particularly the average line coverage across all submissions, is a poor metric to indicate any performance difference. I suggest the authors not justify or argue for anything based on this result. The slightly lower coverage reported for C++ is likely caused by some of the generated tests failing to compile, as mentioned in Section 2.2.
6. The presentation in this paper is not self-consistent. Many statements seem without sufficient support to me. The figure is also misleading, as correct code submissions are given to the LLMs for the problem-solving task as well.

[1]: Mirzadeh, et al., "GSM-Symbolic: Understanding the Limitations of Mathematical Reasoning in Large Language Models" https://arxiv.org/abs/2410.05229

[2]: He, et al., "Evaluating Program Semantics Reasoning with Type Inference in System F" https://arxiv.org/abs/2509.23686

**Questions:**

Please address the concerns raised in **Weaknesses**.

In general, I like the direction of this work, where the authors pay attention to the core nature of generating tests: detecting errors in software. I also like the idea of VMR as a stricter version of pass rate. However, this work requires serious revision to be considered for publication at venues like ICLR.

---

### Official Review · Reviewer_at96 · 2025-10-28

**Soundness:** 2
**Presentation:** 3
**Contribution:** 2
**Rating:** 2
**Confidence:** 4

**Summary:**

The paper introduces TestJudge, a new benchmark to evaluate LLM-based unit test generation. TestJudge focuses on the error correction capabilities of generated tests by leveraging the test judgement for 10 diverse code submissions. The paper further performs an empirical comparison with the proposed benchmark for several LLMs, showing a discrepancy between the  test generation versus direct problem-solving tasks

**Strengths:**

Evaluation of unit test generation is an important problem.

**Weaknesses:**

The novelty of the benchmark needs more justification. The paper claims that the major contribution of this benchmark lies in measuring the capabilities of detecting “subtle logical errors”. Such a claim needs more justification, especially for the definition of “subtle logical erorrs”.  First, existing benchmarks with mutation testing could also represent some logical errors (in fact, mutation testing is a very general framework, and it is flexible to extend the mutation operators. For example, authors could find there is a very extensive and diverse pool of mutation operators in existing mutation testing tools https://pitest.org/quickstart/mutators/) Therefore, it needs more evidence to show what is the subtle logical errors that can uniquely captured by this benchmark. Overwise, the novelty and significance can be restricted.

Second, there is no empirical comparison between the proposed evaluation framework and existing ones. In particular, why could the proposed framework be considered as more accurate for measuring the quality of generated tests? There is no evidence to support the superiority  of the proposed benchmark over existing ones.
The task formulation of test generation is not realistic. As mentioned in line 125, LLMs are given  the problem description, input–output specifications, and input–output samples, ten code submissions with known correctness labels,  for test generation.  Such an input is not always available in practice. In particular, at most cases, only the code for testing can be provided for test generation (as shown in most existing test generation work).


Moreover, the benchmark only focuses on function-level unit test generation, which simplifies the real-world unit test generation scenario too much.  The real-world unit test generation task must face the cross-file/module dependency challenges, and achieving the high coverage for such a scenario is much challenging than function-level unit test generation. Therefore, the benchmark is not representative for the true challenges in unit test generation.

The main findings that the models mainly rely on problem memorization rather than developing robust testing strategies needs more evidence to support. The observation shows that discrepancy between problem resolution and test generation. However, such a discrepancy cannot directly support the claim.

**Questions:**

1.	What is the definition of “subtle logical errors”? Given existing mutation operators already capture a diverse range of logical errors, what kind of errors can be uniquely represented by the proposed benchmark?

2.	Is the task application scenario practical? As problem descriptions and specifications are not always available in practice.

---

### Official Review · Reviewer_8uFT · 2025-10-28

**Soundness:** 2
**Presentation:** 2
**Contribution:** 2
**Rating:** 2
**Confidence:** 4

**Summary:**

This paper presents TestJudge, a benchmark to evaluate the test generation (in functional form) abilities of LLMs for code consisting of codeforces problems in C and python. Each test function is scored based on if it matches the ground truth correctness judgement of 10 submissions for a problem as well as coverage of lines of code executed. The benchmark shows that language models (both frontier and open models) are currently lacking in test generation abilities when compared to code generation abilities with a significant disparity between test generation abilities in python vs. C.

**Strengths:**

- Unit test generation for code is an important area due to its impact on improving code-generation abilities of language models during training and inference, and the use of LLM-as-agents at large, which makes study in this area, and by extension, this work, valuable.
- The benchmark itself could be a test bed for future work to improve upon, so it could be potentially useful, and the paper itself is easy to follow.

**Weaknesses:**

- Missing citations and connections to related work: The paper does not discuss or cite few works in the LLM coding domain that are relevant, while the notion of a test (code vs input-output pairs) is different, they are conceptually similar:
     - https://arxiv.org/abs/2207.10397
     - https://arxiv.org/abs/2502.01619
     - Comparing and contrasting to these works will improve the strength of the paper
- Problem Memorization argument: The authors argue that the disparity in code generation vs test generation performance indicates "problem memorization" which is not convincing enough (see modeling points below in their experimental setting). To me it appears that the task might be out-of-domain for the code models, and it is unclear if simple remedies like 1-shot prompting would improve test generation since most of these models have large context lengths.
- Experimental Setup: Some issues or follow-up on the experimental setting:
     - The authors never fully justify the choice of using test functions, instead of input output pairs when the data is mainly sourced from CodeForces (competitive programming) which often can be represented in both. One advantage of the latter as per https://arxiv.org/abs/2502.01619 is that authors have shown using chain of thought reasoning for generating each unit test is helpful along with remedies like test-time scaling to further improve performance. Some ablations around this would support if the models are truly bad at the broader task of test generation or at doing it all at once, generate multiple unit tests in the same code in an unstructured way.
    - Another line of code-generation works focus on code-debugging, eg. for very simple compilation errors you provide execution feedback to the LLM and ask it to debug its generation (https://arxiv.org/abs/2304.05128). Unclear if that applies or benefits here.
    - Need to test on more open source models like Qwen, Llama, Gemma etc to see how much of a gap persists in open vs frontier models.
    - It appears due to abundance of code-generation data, the same can be used to collect some annotations for test generation (see pipeline described in https://arxiv.org/abs/2502.01619) and used for training open models. It would be interesting to see how much of the gap in performance can we recover from simple SFT on such data.

**Questions:**

See above.

---

### Official Review · Reviewer_mNyH · 2025-10-31

**Soundness:** 3
**Presentation:** 3
**Contribution:** 2
**Rating:** 2
**Confidence:** 4

**Summary:**

This paper proposes a new benchmark for evaluating the test code generation capability of LLMs. The authors start with a data source, Codeforces, and perform filtering to obtain valid sets of problems, each with 10 code submissions (with known pass/fail labels). The main novelty of the benchmark is to evaluate the test code against the 10 submissions to examine if it can correctly assess the pass/fail of the submissions. This is mentioned as Verdict matching rate.
The authors evaluated various LLMs on their benchmark, as well as on the original version of the problem (problem solving), and highlighted gaps in the models' capability to fully understand the logics of the problems vs being able to solve them, potentially thanks to memorization.

**Strengths:**

* The problem setting is sound and the paper is easy to follow.
* Somewhat incremental but novel and strict evaluation of generated test code (VMR).
* The resulting dataset will be useful for the community.

**Weaknesses:**

I think the overall structure of the paper is good, but the proposed novelty, VMR, is not well motivated or explained. Clarity and a more in-depth analysis would strengthen the paper.

* While the number of submissions are investigated in the analysis, it's not clear how they are selected. If the motivation is to measure the subtle logical errors, the selected submissions should reflect the intention. For example, are they diverse? Why would the VMR not drop much when it's >2 submissions, where it's only considered "match" if the verdict for all the submissions should match? It sounds like the selected submissions are somewhat redundant.
* The argued positive correlation between VMR and coverage is questionable. For all the models, in both Python and C++, the coverage scores look like they are in the same range (Python: around 97.7, c++: around 91). Perhaps I am looking at the wrong numbers.
* The absolute gap of percentage between pass@1 and VMR seems to carry less information, because you can change the number of submissions to tweak the range of VMRs.
* Figure 5 makes me wonder how the submissions are selected, because Python "Fail" cases skew towards 1. It seems to make more sense to me to balance the pass / fail classes on average, just like in C++.

**Questions:**

Please see above.

---

### Note · Authors · 2025-12-02

I have read and agree with the venue's withdrawal policy on behalf of myself and my co-authors.